# GC-MS Profiling, Anti-*Helicobacter pylori*, and Anti-Inflammatory Activities of Three Apiaceous Fruits’ Essential Oils

**DOI:** 10.3390/plants11192617

**Published:** 2022-10-05

**Authors:** Hatun A. Alomar, Noha Fathallah, Marwa M. Abdel-Aziz, Taghreed A. Ibrahim, Wafaa M. Elkady

**Affiliations:** 1Pharmacology and Toxicology Department, College of Pharmacy, King Saud University, Riyadh 11451, Saudi Arabia; 2Department of Pharmacognosy and Medicinal Plants, Faculty of Pharmacy, Future University in Egypt, Cairo 11835, Egypt; 3Regional Center for Mycology and Biotechnology (RCMB), Al-Azhar University, Cairo 11651, Egypt; 4Department of Pharmacognosy, College of Pharmacy, King Saud University, Riyadh 11451, Saudi Arabia; 5Department of Pharmacognosy, Faculty of Pharmacy, Cairo University, Cairo 11562, Egypt

**Keywords:** antibacterial, anti-inflammatory, *Carum carvi*, Cuminum cyminum, essential oil, GC-MS *Helicobacter pylori*, molecular docking, Pimpinella anisum

## Abstract

Eradication of *Helicobacter pylori* is a challenge due to rising antibiotic resistance and GIT-related disorders. *Cuminum cyminum, Pimpinella anisum*, and *Carum carvi* are fruits belonging to the Apiaceae family. Their essential oils were extracted, analyzed using GC-MS, tested for anti-*H. pylori* activity by a micro-well dilution technique, identified for potential anti-*H. pylori* inhibitors by an in-silico study, and investigated for anti-inflammatory activity using a COX-2 inhibition assay. Results showed that the main components of *C. cyminum, P. anisum,* and *C. carvi* were cumaldehyde (41.26%), anethole (92.41%), and carvone (51.38%), respectively. Essential oil of *C. cyminum* exhibited the greatest anti-*H. pylori* activity (3.9 µg/mL) followed by *P. anisum* (15.63 µg/mL), while *C. carvi* showed the lowest activity (62.5 µg/mL). The in-silico study showed that cumaldehyde in *C. cyminum* has the best fitting energy to inhibit *H. pylori.*
*C. cyminum* essential oil showed the maximum ability to reduce the production of Cox-2 expression approaching celecoxib with IC_50_ = 1.8 ± 0.41 µg/mL, followed by the *C. carvi* oil IC_50_ = 7.3 ± 0.35 µg/mL and then oil of *P. anisum* IC_50_ = 10.7±0.63 µg/mL. The investigated phytochemicals in this study can be used as potential adjunct therapies with conventional antibiotics against *H. pylori.*

## 1. Introduction

Natural products are believed to have a great role in a wide range of therapeutic conditions. Complementary and alternative medicines are reported to be useful for of many health complaints [1]. Since ancient times, the active substances from plants have been used as remedies for various diseases in the form of hand-made preparations [2]. Several plants belonging to the Apiaceae family were used as spices, flavoring agents, and folk therapies as they have various biological activities [3]. This is because many species in this family contain significant amounts of essential oil as well as beneficial secondary metabolites [4]. In this study, three members are selected (*Cuminum cyminum* L., *Pimpinella anisum* L., and *Carum carvi L.*) that are endogenous to Egypt and the Middle East [5,6,7]. All of them are traditionally used extensively for culinary and medicinal purposes [8,9]. They are recommended for gastrointestinal and neurological complaints and have valuable antibacterial activity. They are also applied in different pharmaceutical, perfume, and cosmetic productions [10]. The three fruits possess essential oils as their main active constituents with recorded antimicrobial activity against various types of microorganisms [9,11,12]. The isolated essential oil of (*C. cyminum* L.) is distinguished by the presence of cuminaldehyde as the major ingredient. It’s commonly used to treat digestive and respiratory problems [13]. *P. anisum* L. is a traditional medicine for bronchitis, cough, colic, nausea, and sleeplessness; moreover, the gastroprotective potential of anise aqueous extract was previously proved [14]. Moreover, (*C. carvi* L.) is also useful for cleaning teeth and treating eczema and pneumonia [15].

*Helicobacter pylori* infection affects more than 50% of the population worldwide. This microorganism has been strongly linked to duodenal and peptic ulcers and different GIT inflammatory disorders, which the WHO categorized as group I carcinogens in 1994 [16]. In Europe and the United States, the common triple therapy of amoxicillin, clarithromycin, and a proton pump inhibitor (PPI) for 7–14 days is the first-line treatment for *H. pylori* infections [17]. However, the effectiveness of traditional triple therapy has declined in the past few years, with eradication rates of fewer than 80% due to the increase in clarithromycin resistance. This highlights the need for new first-line therapies. As a result, the discovery of alternative antimicrobial compounds derived from natural sources is required for improved medicinal uses [18,19,20,21]. 

The current work aims to extract the essential oils of the three chosen fruits and investigate the chemical profile of each oil separately using GC-MS, in addition to evaluating their anti-*H. pylori* and anti-inflammatory activities. Finally, this study also assesses the main constituents identified using in-silico analysis to identify the potential anti-*H. pylori* inhibitors. To our knowledge, this is the first comparison of these three fruits that are indigenous to Egypt against *H. pylori*.

## 2. Results

### 2.1. Essential Oil Extraction

Hydrodistillation of *C. cyminum, P. anisum,* and *C. carvi* fruits vary in their essential oil yields. *C. cyminum* exhibited the highest percentage yield (2.30%), followed by *C. carvi* (1.12%), whereas the *P. anisum* yield was (0.9%). All extracted essential oils had a golden yellowish color with distinct aromatic fragrances. 

### 2.2. GC-MS Analysis of the Essential Oils

The phytochemical profile of the isolated essential oils was distinctly different despite belonging to the same family, Apiaceae. This proposes a potentially wide range of the potency of the biological activities. GC-MS resulted in the identification of 18 different compounds, mainly belonging to the monoterpenes and sesquiterpenes classes. Cuminaldehyde (41.26%), anethole (92.41%)*,* and Carvone (51.38%) were the major constituents in *C. cyminum*, *P. anisum,* and *C. carvi,* respectively (Table 1, Figure 1 and Figure 2). 

### 2.3. In Vitro Evaluation of Anti–H. pylori Activity

Results demonstrated that all examined essential oils have a promising potential against *H. pylori* activity (Table 2). *C. cyminum* showed the greatest activity with Minimum inhibitory concentration (MIC 3.9 µg/mL) comparable to the reference drug; clarithromycin. *P. anisum* showed moderate activity (MIC 15.63 µg/mL), while *C. carvi* showed the lowest activity (MIC 62.5 µg/mL). The variation in the activity could be related to the difference in the essential oil compositions.

### 2.4. Anti-Inflammatory Assay

It was deemed necessary to investigate the potential of the essential oils as anti-inflammatory agents because *H. pylori* infection is frequently linked to many types of gastric inflammations, ranging from gastritis to systemic and vascular inflammation [26,27]. The results were determined by the percentage of inhibition and expressed as mean ± standard deviation, as summarized in Figure 3. Overall, the results indicated that in comparison to the usual medication, celecoxib (positive control) IC_50_ = 0.43 ± 0.12, the three analyzed oils inhibited COX-2 expression but to various levels. As seen in Figure 3, it was noted that the oil of *C. cyminum* showed the strongest activity, approaching celecoxib with IC_50_ = 1.8 ± 0.41, followed by the *C. carvi* oil IC_50_ = 7.3 ± 0.35, and the weakest anti-inflammatory activity was observed in the oil of *P. anisum* IC_5 0_= 10.7 ± 0.63.

### 2.5. In Silico Evaluation of anti-Helicobacter pylori Activity

The strategy of drug discovery and development is searching for herbal medicinal metabolites as ailment inhibitors. Inhibiting quorum sensing, which is essential for bacterial cell proliferation, is an important target for studies aiming to create remedies for bacterial pathogens. It may prevent the formation of drug resistance bacterial strains; this is because preventing the synthesis of virulence factors may allow the human immune system to clear the infection. The 5’-methylthioadenosine/S-adenosylhomocysteine nucleosidase (MTAN) enzyme (3NM4) is vital in bacterial metabolism, especially in diseases where quorum sensing is important [28].

The in-silico study revealed that cumaldehyde, estragole, and *p*-cymene have high fitting scores compared to the co-crystallized ligand (TRIS) (−28.75, −28.72, −28.49 kcal/mol), respectively (Table 3), proposing that they should possess good inhibiting activity against the MTAN enzyme (3NM4). This may be explained by the studied compounds’ capacities to interact with the essential amino acids through H-bonding (Figure 4). Moreover, carvone and anethole have very similar fitting energy as TRIS (−28.22, −28.01, and −28.24, respectively). This could be related to the anti-*H. pylori* activity of the tested essential oil. *C. cyminum* essential oil has the greatest in vitro anti-*H. pylori* activity, which could be related to its main component, cumaldehyde (41.26%). In the same way, anethole, the major constituent in *Pimpinella anisum* (92.41%), could be the cause for its effect. 

## 3. Discussion

*H. pylori*, is a Gram-negative microaerophilic bacterium which affects more than half of the human population worldwide [29]. *H. pylori* infection is the leading cause of gastric and duodenal ulceration and the main risk factor for gastric adenocarcinoma and lymphoma. This infection is usually acquired during early childhood, with higher rates in developing countries [30,31,32]. In Egypt, about 50% of children acquire the infection by 10 years of age [30].

Gastric inflammation, a physiological process that can be triggered by a bacterial infection or tissue injury, is an invariable finding in patients infected with *H. pylori* and appears as an immune response from the host to the organism [33,34]. LPS (lipopolysaccharides) are chemical signals originating from the host and play important roles in the host’s defense against this infection. They activate macrophages to generate high amounts of proinflammatory enzymes such as cyclooxygenase, inducible NO synthase (iNOS), and (COX-2) [27]. *H. pylori* hits various cellular proteins to alter the host inflammatory response, thus initiating multiple attacks on the gastric mucosa, resulting in gastritis and peptic ulceration [35]. 

The current conventional *H. pylori* treatment typically employs a triple-drug regimen with two antibiotics and a proton pump inhibitor (PPI) with a success rate of 80–90% [36]. However, antibiotic resistance is an issue that is provoking an alarming 90% prevalence of *H. pylori* recorded in some African countries [36]. Other major obstacles leading to low eradication rates are the lack of therapeutic compliance, the degradation of antibiotics at gastric pH, and their insufficient residence time in the stomach [32]. Therefore, there is a continuing need for new antimicrobial drugs and creative alternative treatment methods. 

Phytochemicals, which are good for human health and may even help in preventing diseases, are mainly found in plants. Herbal drugs and their essential oils are known to possess a wide range of pharmacological activities [20,37,38]. For thousands of years, essential oils had been used as antibacterial and anti-inflammatory agents [39]. Modern in vitro studies have demonstrated that a variety of bacterial strains are susceptible to the bactericidal effects of essential oils [40,41]. 

Several essential oils were proven to have anti-*H. pylori* potential [37,42,43,44], while others were extensively analyzed for their anti-inflammatory effects and the mechanism of their action was recorded in the literature [45,46,47,48,49,50]. 

This study aimed to emphasize the chemical profile with the investigation of the anti-*H. pylori* and anti-inflammatory activities of the three tested essential oils extracted from *C. cyminum, P. anisum,* and *C. carvi.* They are all indigenous plants of Egypt’s Nile Delta. The chemical profile of each individual oil was obtained by using GC-MS technique. The chromatograms revealed different constituents among the three oils with the presence of numerous oxygenated monoterpenes, such as (limonene and estragole), sesquiterpenes such as (Zingiberene and β-Himachalene), and the monoterpenoid hydrocarbon (β-Pinene). They are known to have powerful antioxidant and antimicrobial activities [51,52,53,54]. Cuminaldehyde, anethole, and carvone were found as the major compounds in the oils of cumin, anise, and caraway, respectively. These compounds may contribute to the anti-*H. pylori* activities of the oils as they are reported to possess bactericidal activity against different types of microorganisms [7,22,55,56,57]. 

The in vitro findings against *H. pylori* noted cumin (*C. cyminum*) oil as the most potent, followed by anise (*P. anisum),* then caraway (*C. carvi*) with MIC values of 3.9 µg/mL, 15.63 µg/mL, and 62.5 µg/mL, respectively. The MIC values were evaluated against the standard drug, clarithromycin (MIC = 1.95 µg/mL). Additionally, the three oils revealed a potential for anti-inflammatory activity. However, they differed in their capacity to inhibit COX-2 expression. *C. cyminum* essential oil showed the highest efficacy (IC_50_ = 1.8 ± 0.41) and most closely resembles the common anti-inflammatory COX-2 inhibitor, celecoxib (IC_50_ = 0.43 ± 0.12). *C. carvi* and *P. anisum* had moderate and weak activity with IC_50_ = 7.3 ± 0.35 and IC_50_ = 10.7 ± 0.63, respectively. The main components of the isolated oils may play a main role in the activity variation. The main component of cumin oil, cuminaldehyde, was formerly considered to be a prominent anti-inflammatory agent, approaching NSAIDS as it develops a dual anti-inflammatory COX–LOX inhibition mechanism [58,59]. Limonene, found in anise oil, acts by regulating the iNOS, COX-2, PGE2, TGF-β, and ERK1/2 signaling pathways [60], and carvone in caraway oil works on LPS-initiated oxidative stress markers and pro-inflammatory cytokines [61]. 

These findings are very important, as scientific reports suggest that *H. pylori* infection represents an additional risk factor for peptic ulcer complications in aspirin/NSAID users, limiting their use for its eradication treatment protocol [62,63]. An in-silico analysis employing a molecular docking approach was carried out to clarify how these compounds affect *H. pylori.* A higher energy score compared to the co-crystallized ligand (TRIS) was recorded for cuminaldehdye, as shown in Table 3 and Figure 4. This may help to explain the cumin oil’s remarkable anti-*H. pylori* activity. The mild action of the caraway and anise essential oils in comparison to cumin may be explained by the fact that (-)-carvone and anethole demonstrated similar inhibitory activity to TRIS. However, further in vivo studies are required to confirm the activity.

## 4. Materials and Methods

### 4.1. Plant Material

The three fruits were purchased from Egypt’s local market in January 2022. They were verified throughout the herbariums of Cairo University’s Faculty of Science and National Research Center (NRC), Cairo, Egypt. Voucher specimens (CM-1, AN-1, and CR-1) were kept in the Herbarium of the Pharmacognosy department, Faculty of Pharmacy, Future University in Egypt.

### 4.2. Extraction of Essential Oils

Dried fruits of *C. cyminum, P. anisum,* and *C. carvi* were hydro-distilled for 5 h according to the method described in the European Pharmacopoeia [64] using a Clevenger apparatus (plant: water ratio 1:3, *w*/*v*). The % yield was calculated using the volume (mL) of essential oil for each 100 g of the studied plant. The oily phases were separated, dried over anhydrous sodium sulfate, and kept for further study.

### 4.3. GC-MS Analysis

The analysis was carried out using the approach proposed by Elkady et al. [65]. GC–MS analysis was performed on Shimadzu GCMS-QP 2010 (Tokyo, Japan) with Rtx-5MS capillary column (30 m × 0.25 mm i.d. × 0.25 µm film thickness) (Restek, USA). The capillary column was coupled to a quadrupole mass spectrometer (SSQ 7000; Thermo-Finnigan, Bremen, Germany). The oven temperature was kept at 45 °C for 2 min (isothermal), programmed to 300 °C at 5 °C/min, and kept constant at 300 °C for 5 min (isothermal); the injector temperature was 250 °C. Helium was utilized as a carrier gas with a constant flow rate set at 1.41 mL/min. Diluted samples (1% *v*/*v*) were injected with a split ratio of 15:1 and the injected volume was 1 μL. The MS parameters were as follows: interface temperature, 280 °C; ion source temperature, 200 °C; EI mode, 70 eV; and scan range, 35–500 amu. Components were identified using retention indices and mass spectra matching with the NIST-11 and Wiley library databases, as well as published data in the literature. The retention indices (RI) were computed in comparison to a homologous sequence of *n*-alkanes (C8-C28) administered under identical circumstances. The overall run duration was about 30 min.

### 4.4. In Vitro Evaluation of anti–H. pylori Activity

Rapid screening of the antibacterial activity of the fruit essential oils was done to confirm their previously reported activity by [66,67,68]. Colorimetric broth micro-dilution method using XTT [2,3-bis(2-methoxy-4-nitro-5-sulfo-phenyl)-2H-tetrazolium- 5-carboxanilide] -reduction assay was adopted to determine the MIC against the reference strain of *H. pylori* ATCC 43504 according to [69,70]. The MIC was specified as the extract concentration that produced a 100% decrease in optical density compared with control growth results. Clarithromycin was used as a standard antibacterial agent. The volatile oils were serially diluted in DMSO, and then 50 μL of each dilution were added to wells in a microtiter plate containing 100 μL TSB. 50 μL of adjusted microbial inoculum (10^6^ cells/mL) were added to each well, and then the microtiter plates were incubated in the dark at 37 °C for 24 h. After incubation, 100 μL of freshly prepared XTT were added and incubated again for 1 h at 37 °C. Colorimetric variation in the XTT assay was measured using a microtiter plate reader at 492 nm. 

### 4.5. Anti-Inflammatory Assay

The samples in the concentration range of 125–0.98 µg/mL were tested to investigate the anti-inflammatory response by inhibiting the COX-2 enzyme. The COX (EC 1.14.99.1) activity was monitored as the result of the N,N,N,N-tetramethyl-*p*-phenylenediamine (TMPD) oxidation reaction with arachidonic acid. This assay was performed according to the previously described method [71,72] with slight modifications. The inhibitory activity was determined by monitoring the absorbance’s increase at 611 nm using a microplate reader (BIOTEK; USA). The inhibitory percentages were calculated according to the formula: Inhibitory activity (%) = (1 − As/Ac) ×100, whereas is the absorbance in the presence of the test substance and Ac is the absorbance of control. The efficacy of the extracts and the reference compound (celecoxib) to inhibit COX-2 isoenzyme was determined as the concentration causing 50% enzyme inhibition (IC_50_).

### 4.6. In Silico Evaluation of anti-H. pylori Activity

The in-silico molecular docking analysis for the discovered compounds in the extracted essential oils in the active center of *H. pylori* MTAN (5′-methylthioadenosine/S-adenosylhomocysteine nucleosidase) enzyme (PDB ID: 3NM4) was performed using Discovery Studio 4.5 (Accelrys, Inc., San Diego, CA, USA). This enzyme is essential for bacterial metabolism. The Protein Data Bank (http://www.pdb.org) accessed 5 May 2022, was used to obtain the protein (enzyme). The co-crystallized ligand tris[hydroxymethyl]aminomethane (TRIS) was used to determine the docking binding site. The free binding energies were computed for the most stable docking locations of the co-crystalized ligand, as well as for all identified molecules [73].

## 5. Conclusions

Cumin, anise, and caraway essential oils are considered rich sources of bioactive compounds. The fruits are widely available and highly consumed due to their nutritional and medicinal value. The current work demonstrates the great anti-*H. pylori* activity of the cumin essential oil due to its major constituents. However, anise and caraway essential oils showed moderate activity. This could offer a natural solution in the pharmaceutical field to combat this bacterial infection. The in-silico study confirmed the activity of the identified constituents. Further in vitro and in vivo studies are required to investigate the synergistic or additive interactions of the oils with the antibiotics to obtain the best results.

## Figures and Tables

**Figure 1 plants-11-02617-f001:**
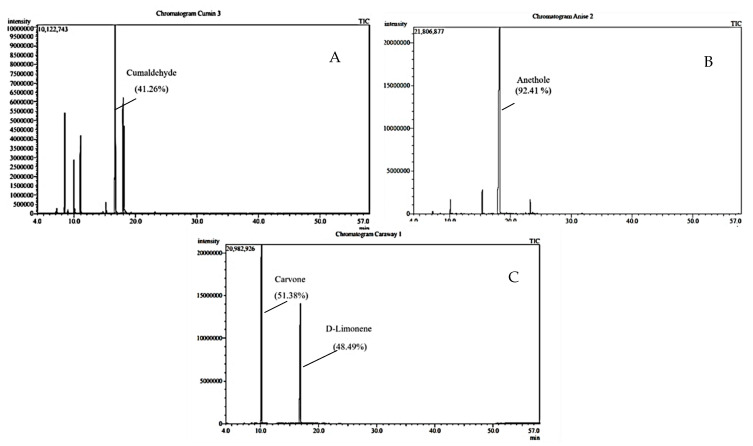
GC chromatograms of the essential oils of (**A**) *C. cyminum*, (**B**) *P. anisum,* and (**C**) *C. carvi*.

**Figure 2 plants-11-02617-f002:**
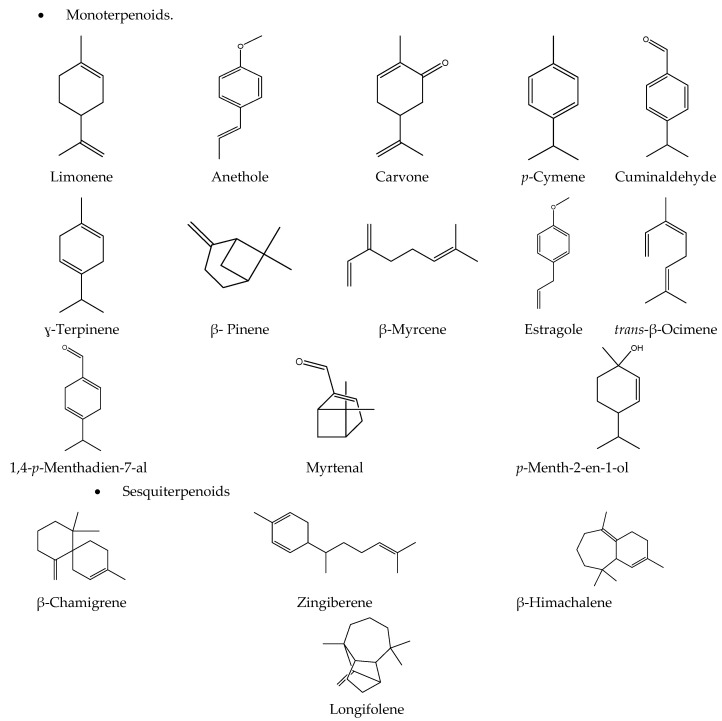
2D structures of the main detected compounds of three oils L. by GC–MS analysis (Drawn by Chem-draw ultra-version 14).

**Figure 3 plants-11-02617-f003:**
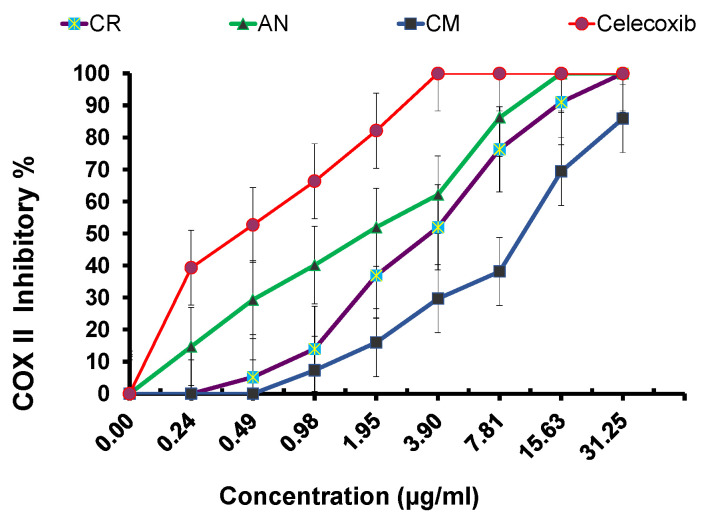
Anti-inflammatory activity cyclooxygenase COX-2 inhibitory % of CM (*C. cyminum*), AN (*P. anisum*), and CR (*C. carvi*) compared to standard drug, celecoxib.

**Figure 4 plants-11-02617-f004:**
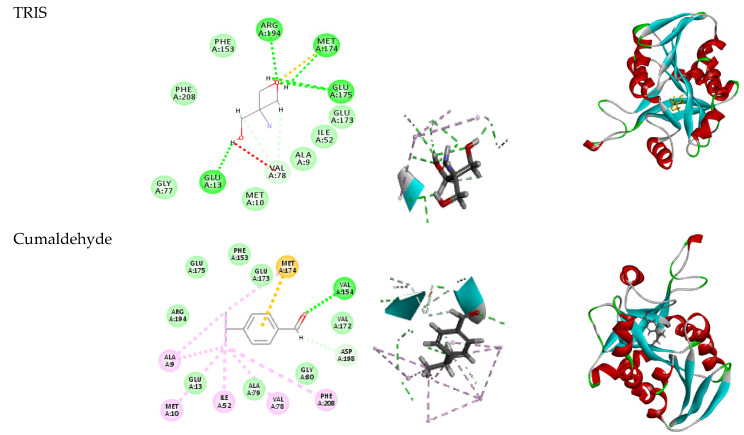
2D and 3D interaction diagram of the co-crystallized ligand and identified compounds docking pose interactions with the key amino acids in the 3NM4 binding site. The three-letter amino acid codes and positions of each residue are indicated. Between the receptor and the ligand, hydrogen-bonding interactions are illustrated by a green dashed line, whereas -alkyl interactions are represented by a purple dashed line.

**Table 1 plants-11-02617-t001:** Chemical compositions of the isolated essential oils.

Compound	RI _exp_	RI _rep_	*C. cyminum*	*P. anisum*	*C. carvi*	References
1- Monoterpenoids						
Cumin aldehyde	1283	1283	41.26	n.d	n.d	[22]
Limonene	1027	1027	0.77	1.57	48.49	[3,23]
Carvone	1244	1249	n.d	n.d	51.38	[23]
Anethole	918	918	n.d	92.41	n.d	[3]
ɣ-Terpinene	1049	1049	8.11	n.d	n.d	[22]
1,4-*p*-Menthadien-7-al	1784	1784	12.84	n.d	n.d	[23]
Myrtenal	1237	1237	19.26	n.d	n.d	[23]
α-Pinene	933	930	0.43	n.d	n.d	[22]
β-Pinene	969	973	10.27	n.d	n.d	[22]
β-Myrcene	980	985	0.41	n.d	0.13	[22]
*p*-Cymene	1441	1441	5.22	n.d	n.d	[22]
Estragole	1197	1197	n.d	3.01	n.d	
*trans*-β-Ocimene	1034	1034	n.d	0.22	n.d	[3]
*p*-Menth-2-en-1-ol	1130	1129	1.26	n.d	n.d	[22]
2- Sesquiterpenoids						
β-Chamigrene	1478	1478	n.d	0.20	n.d	[24]
Zingiberene	1494	1494	n.d	0.26	n.d	[25]
β-Himachalene	1505	1505	n.d	0.10	n.d	[25]
Longifolene	1532	1532	n.d	2.11	n.d	[25]
% of identification			99.83	97.21	100	

All compounds were identified by Kováts index and mass spectral data by comparing values reported in the literature. **RI _exp_** Kovats index was determined experimentally relative to C8–C28 n-alkanes. **RI _rep_** reported Kovats retention indices.

**Table 2 plants-11-02617-t002:** Mean of *H. pylori* inhibitory % of three extracted essential oils against the standard drug clarithromycin.

Sample conc. (µg/mL)	Mean of *H. pylori* Inhibitory%			
	Clarithromycin *	*C. cyminum **	*P. anisum **	*C. carvi **
125	100	100.00	100.00	100.00
62.5	100	100.00	100.00	100.00
31.25	100	100.00	100.00	81.63 ± 1.3
15.63	100	100.00	100.00	70.64 ± 2.1
7.81	100	100.00	79.21 ± 0.5	46.21 ± 0.59
3.9	100	100.00	51.74 ± 1.3	21.74 ± 1.3
1.95	100	81.97 ± 0.95	29.39 ± 0.6	8.29 ± 0.74
0.98	92.45 ± 1.2	73.25 ± 1.3	17.92 ± 1.5	3.71 ± 0.91
0.48	87.65 ± 0.5	59.31 ± 1.6	6.34 ± 0.74	0.00
0.24	81.35 ± 1.5	27.28 ± 1.0	0.00	0.00
0.00	0.00	0.00	0.00	0.00
MIC	1.95	3.9	15.63	62.5

* All determinations were carried out in a triplicate manner and values are expressed as the mean ± SD.

**Table 3 plants-11-02617-t003:** Free binding energies (∆G) of the identified compounds within the 3NM4 active site calculated in kcal/mol using Discovery Studio 4.5, adopting both rule-based ionization techniques.

Compound	Binding Energy ∆G (Kcal/mol) Rule-Based
Cumaldehyde	−28.75
Estragole	−28.72
*p*-Cymene	−28.49
TRIS	−28.24
(-)-Carvone	−28.22
Anethole	−28.01
ɣ-terpinene	−25.32
Limonene	−25.26
α Thujenal	−22.32
β Pinene	−21.88
Myrtenal	−20.80

## Data Availability

Not applicable.

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
