# Peer review of "GC-MS Profiling, Anti-Helicobacter pylori, and Anti-Inflammatory Activities of Three Apiaceous Fruits’ Essential Oils"

_plants, 2022, doi:10.3390/plants11192617_

Round 1

Reviewer 1 Report

Please refer to the observations/suggestions available as notes in the enclosed file.

Author Response

Thank you for your consideration of our manuscript (plants-1936684)
entitled "GC/MS Profiling, Anti-Helicobacter pylori, and Anti-inflammatory activities of three Apiaceous fruits essential oil". Your comments and those of the reviewers were highly insightful and enabled us to greatly improve the quality of the manuscript. We do our best to respond point-by-point to each of the comments of the reviewers.

Revisions in the text are shown as; yellow highlights are the responses for reviewer 1, the blue highlights are the responses for reviewer 2, and the green highlights are the responses for reviewer 3.

We hope that the revisions in the manuscript and our accompanying responses will be sufficient to make our manuscript suitable for publication in Plants.

Response to Reviewer 1 Comments

Point 1: Please refer to the observations/suggestions available as notes in the enclosed file.

Response 1:

We would like to express our deep thanks for the reviewers` opinion and we hope that we can fit all the required corrections; all required modifications were yellow highlighted.

Reviewer 2 Report

The paper is about GC/MS Profiling, Anti- Helicobacter pylori, and Anti-inflammatory activities of three Apiaceous fruits essential oil. Please find my suggestions:

Please check the Instructions for authors regarding the length of the abstract.

L44. After the reference [1] it should be added that "since ancient times, the active substances from plants have been used as remedies for various diseases, in the form of hand-made preparations [Bungau S.G., Popa V.-C. Between religion and science: some aspects: concerning illness and healing in antiquity, Transylv. Rev., 26(3), 2015, 3-19].”

L70-74. Please make the aim of the study a separate, last paragraph of the Introduction. Furthermore, please highlight better the novelty of your research.

Remove empty lines in the manuscript, e.g. L99, 100, 123, 138, 224, 235, etc. They give a sloppy look to the manuscript. Check the entire manuscript in this regard.

Discussion section. Please divide it in more paragraphs, according to ideas they develop. It is much too compact.

After L223, please add in a separate paragraph the strengths and the weakness of your research.

ml must be corrected as mL, as Litter being the international unit of measure for volume. Revise the entire manuscript in this regard, including the tables.

Author Response

Thank you for your consideration of our manuscript (plants-1936684)
entitled "GC/MS Profiling, Anti-Helicobacter pylori, and Anti-inflammatory activities of three Apiaceous fruits essential oil". Your comments and those of the reviewers were highly insightful and enabled us to greatly improve the quality of the manuscript. We do our best to respond point-by-point to each of the comments of the reviewers.

Revisions in the text are shown as; yellow highlights are the responses for reviewer 1, the blue highlights are the responses for reviewer 2, and the green highlights are the responses for reviewer 3.

We hope that the revisions in the manuscript and our accompanying responses will be sufficient to make our manuscript suitable for publication in Plants.

Response to Reviewer 2 Comments

The paper is about GC/MS Profiling, Anti- Helicobacter pylori, and Anti-inflammatory activities of three Apiaceous fruits essential oil. Please find my suggestions:

Point 1: Please check the Instructions for authors regarding the length of the abstract.

We would like to express our gratitude for the reviewer's input. The abstract is summarized as recommended.

Point 2: L44. After the reference [1] it should be added that "since ancient times, the active substances from plants have been used as remedies for various diseases, in the form of hand-made preparations [Bungau S.G., Popa V.-C. Between religion and science: some aspects: concerning illness and healing in antiquity, Transylv. Rev., 26(3), 2015, 3-19].”

We thank the reviewer for such a valuable remark, the references is added.

Point 3: L70-74. Please make the aim of the study a separate, last paragraph of the Introduction. Furthermore, please highlight better the novelty of your research.

We appreciate the reviewer's thoughtful comment; please check the modification done as recommended (blue highlights in the introduction section).

Point 4: Remove empty lines in the manuscript, e.g. L99, 100, 123, 138, 224, 235, etc. They give a sloppy look to the manuscript. Check the entire manuscript in this regard.

We apologize for this sloppy look and your respected comment is considered.

Point 5: Discussion section. Please divide it in more paragraphs, according to ideas they develop. It is much too compact.

Done.

Point 6: After L223, please add in a separate paragraph the strengths and the weakness of your research.

We appreciate the reviewer's helpful comment; please check the modification done as recommended (blue highlights in the discussion section).

Point 7: ml must be corrected as mL, as Litter being the international unit of measure for volume. Revise the entire manuscript in this regard, including the tables.

We apologize for this typographic error; the entire manuscript is revised in this regard

Reviewer 3 Report

Eradication of Helicobacter pylori is one of the world's most important health challenges. This study was represented by GC/MS Profiling after refining essential oils, and then tested their effects on Anti-Helicobacter pylori, and An-2 ti-inflammatory activities. This research has important implications for the development of plant medicines. The paper is worth publishing. But there are still many problems in the writing of the paper, and the author is expected to correct them.

1、 keywords, generally requiring alphabetical orderly;

2、 When a botanical name first appears, write the full name. The second appearance is abbreviated as Lines 5, 59, 148;

3、 The basic format needs to pay attention to Line 4, 48, 61, 72, 143;

4、 Figure 1, this figure can be used or not. If used, you must be repeated several times, with significant differences, Lines 81-83;

5、 Table1, a table listing the names of different compounds, should be the result of previous research. It should be supplemented in the MM line 48-249 previous literature, with only one GC/MS profiling cannot obtain a specific compound name, it must be operated by single compound chromatography and mass spectrometry analysis to obtain;

6、 Figure 3, the figure can be corrected a little, reducing the entire figure space; please refer to Mnif and Aifa 2015.

7、 Table 3 and Figure 4 have the same content, it is recommended to delete one of them, and it is recommended to retain Figure 4;

8、 Figure 5 takes up a lot of space in the text, but it is useless, it is recommended to delete Figure 5, we have Figure 3.

Author Response

Thank you for your consideration of our manuscript (plants-1936684)
entitled "GC/MS Profiling, Anti-Helicobacter pylori, and Anti-inflammatory activities of three Apiaceous fruits essential oil". Your comments and those of the reviewers were highly insightful and enabled us to greatly improve the quality of the manuscript. We do our best to respond point-by-point to each of the comments of the reviewers.

Revisions in the text are shown as; yellow highlights are the responses for reviewer 1, the blue highlights are the responses for reviewer 2, and the green highlights are the responses for reviewer 3.

We hope that the revisions in the manuscript and our accompanying responses will be sufficient to make our manuscript suitable for publication in Plants.

We shall look forward to hearing from you at your earliest convenience.

Response to Reviewer 3 Comments

Eradication of Helicobacter pylori is one of the world's most important health challenges. This study was represented by GC/MS Profiling after refining essential oils, and then tested their effects on Anti-Helicobacter pylori, and Anti-inflammatory activities. This research has important implications for the development of plant medicines. The paper is worth publishing. But there are still many problems in the writing of the paper, and the author is expected to correct them.

We would like to express our heartfelt gratitude for the reviewers' feedback, and we hope that we can incorporate all the necessary changes.

1 keywords, generally requiring alphabetical orderly;

We express regret for this typographical error. Corrected

2 When a botanical name first appears, write the full name. The second appearance is abbreviated as Lines 5, 59, 148;

Thank you for the valuable remark. Corrected

3 The basic format needs to pay attention to Line 4, 48, 61, 72, 143;

We appreciated this insightful feedback. Done

4 Figure 1, this figure can be used or not. If used, you must be repeated several times, with significant differences, Lines 81-83;

The figure is deleted as recommended

5 Table1, a table listing the names of different compounds, should be the result of previous research. It should be supplemented in the MM line 48-249 previous literature, with only one GC/MS profiling cannot obtain a specific compound name, it must be operated by single compound chromatography and mass spectrometry analysis to obtain;

We appreciate the reviewer's thoughtful comment; the method of identification of the obtained compounds is (green highlighted) in the materials and methods section, moreover, references were added in table (1) as the confirmation of the compounds` identity was done through comparison with previously published data.

6 Figure 3, the figure can be corrected a little, reducing the entire figure space; please refer to Mnif and Aifa 2015.

The figure is adjusted as recommended and the reference is added

7 Table 3 and Figure 4 have the same content, it is recommended to delete one of them, and it is recommended to retain Figure 4;

The table is deleted as recommended

8 Figure 5 takes up a lot of space in the text, but it is useless, it is recommended to delete Figure 5, we have Figure 3.

Figure 5 represents the in-silico study of the major identified component in the essential oil while Figure 3 just represents the 2D structures of the main detected compounds “number of figures in the manuscript is now changed after the applied modifications”. We apologize that we couldn`t delete it because it represents one of the main results required for the aim of this work [evaluating the main constituents in each oil using in-silico analysis to identify the potential anti-H. pylori inhibitors]. Thanks for your kind understanding and accepting

Round 2

Reviewer 3 Report

Since the authors have been revised the manuscript, it is accepted after a mini correction with the Tables and figures, which should be arranged in one page. (Fig.2,4, and Table 1, 3).

Author Response

Response to Reviewer 3 Comment

Since the authors have been revised the manuscript, it is accepted after a mini correction with the Tables and figures, which should be arranged in one page. (Fig.2,4, and Table 1, 3).

We would like to express our deep thanks for the reviewers` opinion, and we hope that we can fit all the required corrections

We appreciate the reviewer's thoughtful comment; tables and figures were adjusted in one page as recommended, but we apologize about figure 4 as it has many rows, so we can`t fitted in a single page.
